# Federated Learning of Sparse Gaussian Processes

## Abstract

Gaussian processes (GPs) are widely used flexible nonparametric probabilistic models, and sparse variational approximations for GPs (sparse GPs) have emerged as the go-to method for addressing their poor computational efficiency. In many applications in which we would like to use sparse GPs, datasets are distributed across multiple clients and data privacy is often a concern. This motivates the use of federated learning algorithms, which enable clients to train a model collaboratively without centralising data. Partitioned variational inference (PVI) is an established framework for communication-efficient federated learning of variational approximations. However, we show that PVI cannot support sparse GPs due to the need to share and learn variational parameters (the inducing point locations) across clients. Hence, we re-frame inducing points in sparse GPs as auxiliary variables in a hierarchical variational model (HVM). We use this reformulation to extend PVI to variational distributions with shared variational parameters across client-specific factors, enabling communication-efficient federated learning of inducing points. In addition, we develop a novel parameterisation of the variational distribution which, when combined with the HVM formulation of inducing points, improves the communication efficiency and quality of learning. Our experiments show that our method significantly outperforms baseline approaches for federated learning of sparse GPs on a number of real-world regression tasks.

## 1 Introduction

Gaussian processes (GPs) provide a flexible and robust nonparametric Bayesian approach to modelling complex data patterns, and sparse variational approximations for GPs (sparse GPs) have enabled their deployment on large datasets and arbitrary likelihood functions (Rasmussen & Williams, 2006; Titsias, 2009; Hensman et al., 2013). This makes them an attractive choice for practitioners in a host of problem settings, including regression, classification and time-series forecasting. Yet, in many applications in which we would like to deploy sparse GPs—such as healthcare and spatio-temporal modelling—data are distributed across multiple clients and data privacy is often a concern. Centralising data in such settings is infeasible, preventing the deployment of standard learning algorithms such as variational inference (VI). Instead, we must resort to the use of federated learning algorithms, which allow multiple clients to collaboratively train a machine learning model whilst retaining their data locally (McMahan et al., 2017; Kairouz et al., 2021). Partitioned variational inference (PVI) (Ashman et al., 2022) has emerged as a framework for performing VI in the federated learning setting, offering advantages over more general frameworks such as FedAVG (McMahan et al., 2017). PVI operates by decomposing the variational approximation into factors, each of which approximates the contribution of a single client's data to the posterior. This natural decomposition leads to faster convergence than applying more generic methods.

Whilst PVI has been deployed on variational approximations for a range of probabilistic models, sparse GPs have remained elusive. At the crux of this is the incompatibility of these approximations with PVI. In order to support federated sparse GPs, approximations that share inducing points across clients are needed. However, whilst PVI supports federated learning of *model parameters* that are shared across clients, it does not support federated learning of *variational parameters* that are shared across clients. This is because PVI explicitly tracks the contributions to the posterior from each client to the posterior over model parameters—it does not have a way representing the contributions to shared variational parameters. So, if we naïvely

apply PVI, each client-level optimisation of the inducing points leads to them being fit to the client's data and forgetting contributions from all other clients. Our key insight is that if a distributional estimate was used instead, then PVI could be used to update each client's contributions to this distribution in an identical manner to how it updates clients' contributions to the approximate posterior. In this paper, we achieve this by combining PVI with the framework of hierarchical variational models (HVMs) (Ranganath et al., 2016) to enable its deployment on approximations with shared variational parameters. Similar to the standard PVI approach, each client maintains a contribution to this distribution in the form of a factor, which can be easily combined if these factors are restricted to exponential families. Another crucial step in the application of PVI to sparse GPs is that of a novel parameterisation of the variational distribution, which decouples the mean and covariance of the approximate posterior from the choice of inducing locations. Doing so both improves the communication-efficiency of federated learning and the quality of the approximate posterior when inducing locations are stochastic. Although our main motivation is federated learning of sparse GPs, we additionally show how our framework can be used to perform federated learning of general variational approximations that have shared variational parameters through application to variational autoencoders (VAEs) (Kingma & Welling, 2013). We highlight the following contributions:

1. **A framework for performing PVI with shared variational parameters:** We extend the applicability of PVI to approximations with shared variational parameters using HVMs. We show that by using HVMs to slightly worsen the variational approximation in terms of the ELBO, we can maintain distributional estimates over the auxiliary variables and apply PVI.

2. **A novel parameterisation of sparse GPs:** We develop the decoupled pseudo-observation (DPO) variational approximation for sparse GPs, which generalises existing parameterisations in a manner which improves both the communication-efficiency of federated learning, and quality of the approximate posterior when inducing locations are stochastic.

3. **Empirical evaluation on real-world datasets:** Using multiple federated learning experiments with real-world data, we compare the performance of our method to a several baselines including global VI. The results demonstrate the efficacy of the proposed approach.

## 2 Related Work

**Federated learning of GPs**  As it stands, we are not aware of any existing approaches that perform federated learning of sparse GP variational approximations using PVI. Federated learning of sparse GP approximations using federated averaging (McMahan et al., 2017) is performed by Guo et al. (2022). The use of federated averaging instead of PVI does not take advantage of the structure of the variational approximation. This results in poor communication efficiency.[1] A closely related setting to federated learning is that of distributed learning, in which the objective of scalability is achieved by distributing computation across devices. Notably, Gal et al. (2014) perform distributed learning of sparse GPs by noting that the optimal variational approximation—given the inducing locations and hyperparameters—can be expressed as a summation over data points and thus can be performed in parallel. To learn the inducing locations and hyperparameters, they iterate between performing gradient-based updates on the server side and computing the optimal approximations given the updated values on the client side. This is similar to the hyperparameter optimisation scheme proposed by Ashman et al. (2022), and suffers from poor communication efficiency. Deisenroth & Ng (2015) also develop a method for distributed approximate inference in GPs based on the Bayesian committee machine. Similar to Gal et al. (2014), their method is unsuitable for the federated learning setting as it requires access to the entire dataset when making predictions. Finally, Achituve et al. (2021) perform federated learning of a shared neural network which they use to extract features, which in turn are used by GPs local to each client. They do not learn a global GP approximation, and thus focus on the personalised federated learning setting.

---

[1]Throughout, we refer to communication efficiency as the number of local optimisation procedures performed until convergence of the variational approximation.

**Bayesian FITC** The method we use is related to the Bayesian FITC approximation of Rossi et al. (2021), which also involves probabilistic treatment of inducing locations but is otherwise quite different. We defer discussion to Section 5.2, after introducing our method.

## 3 Partitioned Variational Inference

Partitioned variational inference (PVI) (Ashman et al., 2022) extends VI to the federated learning setting, and thus serves as an attractive foundation for federated learning of sparse GPs. Consider the task of modelling data on $K$ clients, $\mathbf{y} = \{\mathbf{y}_k \in \mathcal{Y}^{N_k}\}_{n=1}^{N_k}$ [2], with a probabilistic model defined by the joint distribution $p(\mathbf{y}, \boldsymbol{\theta}) = p(\boldsymbol{\theta}) \prod_{k=1}^{K} p(\mathbf{y}_k|\boldsymbol{\theta})$, where $\boldsymbol{\theta}$ describes the model and is the variable of inferential interest. In general, exact Bayesian inference is intractable and we resort to seeking a member of a family of tractable distributions $\mathcal{Q}$ which minimises the KL-divergence between itself and the true posterior distribution, or equivalently which maximises the variational lower-bound to the marginal likelihood, $\mathcal{L}$:

$$q^*(\boldsymbol{\theta}) = \arg\max_{q(\boldsymbol{\theta}) \in \mathcal{Q}} \underbrace{\mathbb{E}_q \left[\log p(\mathbf{y}|\boldsymbol{\theta})\right] - \mathrm{KL}\left[q(\boldsymbol{\theta})||p(\boldsymbol{\theta})\right]}_{\mathcal{L}}. \tag{1}$$

Rather than considering a single, global distribution, PVI decomposes the variational approximation in a manner that mirrors the decomposition of the true posterior: $q(\boldsymbol{\theta}) \propto p(\boldsymbol{\theta}) \prod_{k=1}^{K} t_k(\boldsymbol{\theta})$. Here each $t_k(\boldsymbol{\theta})$ factor in the variational approximation corresponds to each $p(\mathbf{y}_k|\boldsymbol{\theta})$ factor in the true posterior, and thus if correctly specified involves only data $\mathbf{y}_k$. Since the factors are approximate and not correctly specified, PVI iteratively refines them through client-level optimisation involving only local data. PVI iterates three steps until convergence:

**①** At each iteration $i$, a central server selects a set of factors $\{t_m^{(i-1)}(\boldsymbol{\theta})\}_m$.

**②** Each factor $\{t_m(\boldsymbol{\theta})\}_m$ is updated by solving $q_m^{(i)}(\boldsymbol{\theta}) = \arg\max_{q(\boldsymbol{\theta}) \in \mathcal{Q}} \mathcal{L}_m^{(i)}$ where

$$\mathcal{L}_m^{(i)} = \mathbb{E}_q\left[\log p(\mathbf{y}_m|\boldsymbol{\theta})\right] - \mathrm{KL}\left[q(\boldsymbol{\theta})||q_{\backslash m}^{(i-1)}(\boldsymbol{\theta})\right]. \tag{2}$$

$q_{\backslash m}^{(i-1)}(\boldsymbol{\theta}) \propto \frac{q^{(i-1)}(\boldsymbol{\theta})}{t_m^{(i-1)}(\boldsymbol{\theta})}$ is termed the cavity distribution and $\mathcal{L}_m^{(i)}$ is referred to as the local (negative) free-energy for client $m$ at iteration $i$. Given $q_m^{(i)}(\boldsymbol{\theta})$, the updated approximate likelihood is found by division, $t_m^{(i)}(\boldsymbol{\theta}) \propto \frac{q_m^{(i)}(\boldsymbol{\theta})}{q_{\backslash m}^{(i-1)}(\boldsymbol{\theta})}$. $\Delta_m^{(i)}(\boldsymbol{\theta}) \propto \frac{t_m^{(i)}(\boldsymbol{\theta})}{t_m^{(i-1)}(\boldsymbol{\theta})}$ is communicated back to the server.

**③** Finally, the approximate posterior is updated as $q^{(i)}(\boldsymbol{\theta}) \propto q^{(i-1)}(\boldsymbol{\theta}) \prod_{m \in b_i} \Delta_m^{(i)}(\boldsymbol{\theta})$.

PVI enjoys several beneficial properties, most notably that if PVI converges then it does so to the global VI solution.

## 4 PVI with Shared Variational Parameters

In this section we propose our new framework, which extends the application of PVI to variational approximations in which some variational parameters, $\boldsymbol{\phi}$, are shared across factors:

$$q(\boldsymbol{\theta}|\boldsymbol{\phi}) = p(\boldsymbol{\theta}) \prod_{k=1}^{K} t_k(\boldsymbol{\theta}|\boldsymbol{\phi}). \tag{3}$$

Unlike variational parameters that are factor-specific, optimisation of shared variational parameters cannot be performed locally by each client as they are treated as free-form parameters to refine $t_k(\boldsymbol{\theta}|\boldsymbol{\phi})$, rather than the global distribution $q(\boldsymbol{\theta}|\boldsymbol{\phi})$, and so would iterate between the set of solutions $\{\boldsymbol{\phi}_k^* = \arg\max_{\boldsymbol{\phi}} \mathcal{L}_k\}_k$

---

[2]We have neglected potential dependence on a set of inputs for notational simplicity.

rather than converge to the global VI solution $\phi^* = \arg\max_\phi \mathcal{L}$. One option is to perform nested federated averaging to update $\phi$ in tandem with PVI updating the factors, as described in Algorithm 1. We show in Section 6 that this method often performs poorly.

Consider, instead, treating $\phi$ as an auxiliary variable in the model

$$q(\boldsymbol{\theta}) = p(\boldsymbol{\theta}) \int \prod_{k=1}^K \overbrace{t_k(\boldsymbol{\theta}|\boldsymbol{\phi})t_k(\boldsymbol{\phi})}^{t_k(\boldsymbol{\theta},\boldsymbol{\phi})} \, d\boldsymbol{\phi}. \tag{4}$$

This is an instance of a hierarchical variational model (HVM) (Ranganath et al., 2016). Rather than involving a fixed-point estimate for $\phi$, this formulation maintains a distribution $q(\boldsymbol{\phi}) = \prod_{k=1}^K t_k(\boldsymbol{\phi})$ which is then integrated out. This allows us to update and combine individual clients' contributions to $\phi$ in a communication efficient manner using PVI. The resulting variational lower-bound is given by

$$\tilde{\mathcal{L}} = \mathbb{E}_q \left[ \log p(\mathbf{y}|\boldsymbol{\theta}) - \mathrm{KL}\left[q(\boldsymbol{\theta}|\boldsymbol{\phi})||p(\boldsymbol{\theta})\right] - \mathrm{KL}\left[q(\boldsymbol{\phi})||r(\boldsymbol{\phi}|\boldsymbol{\theta})\right] \right] \tag{5}$$

where $r(\boldsymbol{\phi}|\boldsymbol{\theta})$ is an auxiliary likelihood used to construct the variational lower-bound which can be freely optimised such that this bound is maximised. The maximum is achieved at $r(\boldsymbol{\phi}|\boldsymbol{\theta}) = q(\boldsymbol{\phi}|\boldsymbol{\theta})$ which, similar to the intractability of $p(\boldsymbol{\theta}|\mathbf{y})$, is generally intractable itself (Ranganath et al., 2016). In the following section, we discuss our choice of $r(\boldsymbol{\phi}|\boldsymbol{\theta})$.

### 4.1 Defining the Auxiliary Likelihood

For the models that we are interested in, there is no clear method for incorporating the dependency of $\phi$ on $\theta$ in the auxiliary likelihood. However, as we show in Appendix A, if we construct $q(\boldsymbol{\phi}) = \tilde{p}(\boldsymbol{\phi}) \prod_{k=1}^K t_k(\boldsymbol{\phi})$, then choosing $r(\boldsymbol{\phi}) = \tilde{p}(\boldsymbol{\phi})$ gives

$$\tilde{\mathcal{L}} = \mathbb{E}_q \left[ \log p(\mathbf{y}|\boldsymbol{\theta}) - \mathrm{KL}\left[q(\boldsymbol{\theta}|\boldsymbol{\phi})||p(\boldsymbol{\theta})\right] \right] - \mathrm{KL}\left[q(\boldsymbol{\phi})||\tilde{p}(\boldsymbol{\phi})\right]. \tag{6}$$

This is identical to the standard variational lower-bound obtained through treating $\phi$ as parameters of the probabilistic model with prior $\tilde{p}(\boldsymbol{\phi})$. PVI can be deployed to target the variational lower-bound in Equation (6) using the local free-energy

$$\tilde{\mathcal{L}}_k = \mathbb{E}_q \left[ \log p(\mathbf{y}_k|\boldsymbol{\theta}) - \mathrm{KL}\left[q(\boldsymbol{\theta}|\boldsymbol{\phi})||q_{\backslash k}(\boldsymbol{\theta}|\boldsymbol{\phi}))\right] \right] - \mathrm{KL}\left[q(\boldsymbol{\phi})||q_{\backslash k}(\boldsymbol{\phi})\right] \tag{7}$$

where $q_{\backslash k}(\boldsymbol{\theta}|\boldsymbol{\phi}) \propto p(\boldsymbol{\theta}|\boldsymbol{\phi}) \prod_{m \neq k} t_m(\boldsymbol{\theta}|\boldsymbol{\phi})$ and $q_{\backslash k}(\boldsymbol{\phi}) \propto \tilde{p}(\boldsymbol{\phi}) \prod_{m \neq k} t_m(\boldsymbol{\phi})$. Pseudo-code for the implementation of sequential PVI using this local free-energy is provided in Algorithm 2. Note that the additional KL term is used solely for ensuring that $q(\boldsymbol{\phi})$ remains stochastic, and so we might consider the effect of its scaling using $\alpha \in [0, 1)$ such that $\tilde{\mathcal{L}}_{k,\alpha} = \mathbb{E}_q \left[ \log p(\mathbf{y}_k|\boldsymbol{\theta}) - \mathrm{KL}\left[q(\boldsymbol{\theta}|\boldsymbol{\phi})||q_{\backslash k}(\boldsymbol{\theta}|\boldsymbol{\phi}))\right] \right] - \alpha\mathrm{KL}\left[q(\boldsymbol{\phi})||q_{\backslash k}(\boldsymbol{\phi})\right]$. We show in Appendix B that only for $\alpha = 1$ are the fixed points of PVI the same as the fixed-points of global VI applied to the variational lower-bound in Equation (6) (with an equivalent scaling of the latter KL term). However, we find that in practice the use of $\alpha < 1$ results in improved performance of the learnt variational approximation. This resembles the performance improvements observed with posterior tempering (Wenzel et al., 2020).

**Algorithm 1:** Sequential PVI using FedAvg to optimise the shared variational parameters.

1: Choose $J$.
2: Initialise $q^0(\boldsymbol{\theta}|\boldsymbol{\phi}) = p(\boldsymbol{\theta}) \prod_{k=1}^{K} t_k^0(\boldsymbol{\theta}|\boldsymbol{\phi})$
3: **for** $i = 1, 2, \ldots$ until convergence **do**
4:     $k :=$ index of next factor to refine.
5:     $q_{k,0}^{(i)}(\boldsymbol{\theta}) \leftarrow \arg\max_q \mathcal{L}_k(q(\boldsymbol{\theta}, \boldsymbol{\phi}))$.
6:     $\boldsymbol{\phi}_{k,0}^{(i)} = \boldsymbol{\phi}^{(i-1)}$.
7:     **for** $j = 1, 2, \ldots, J$ **do**
8:         $q_{k,j}^{(i)}, \boldsymbol{\phi}_{k,j}^{(i)} \leftarrow \nabla\mathcal{L}_k\left(q_{k,j-1}^{(i)}, \boldsymbol{\phi}_{k,j-1}^{(i)}\right)$
9:     **end for**
10:   $t_k^{(i)}(\boldsymbol{\theta}) \propto q_{k,J}^{(i)}(\boldsymbol{\theta})/q_{\backslash k}^{(i-1)}(\boldsymbol{\theta})$
11:   $q^{(i)}(\boldsymbol{\theta}) \propto q_{\backslash k}^{(i-1)}(\boldsymbol{\theta})t_k^{(i)}(\boldsymbol{\theta})$
12:   $\boldsymbol{\phi}^{(i)} = \frac{N-N_k}{N}\boldsymbol{\phi}^{(i-1)} + \frac{N_k}{N}\boldsymbol{\phi}_{k,J}^{(i)}$.
13: **end for**

**Algorithm 2:** Sequential PVI using a HVM to maintain a distribution over the shared variational parameters.

1: Choose $\alpha \in [0, 1]$ and $\tilde{p}(\boldsymbol{\phi})$.
2: Initialise $q^0(\boldsymbol{\theta}, \boldsymbol{\phi}) = p(\boldsymbol{\theta})\tilde{p}(\boldsymbol{\phi}) \prod_{k=1}^{K} t_k^0(\boldsymbol{\theta}, \boldsymbol{\phi})$.
3: **for** $i = 1, 2, \ldots$ until convergence **do**
4:     $k :=$ index of next factor to refine.
5:     $q_k^{(i)}(\boldsymbol{\theta}, \boldsymbol{\phi}) \leftarrow \arg\max_q \tilde{\mathcal{L}}_{k,\alpha}(q(\boldsymbol{\theta}, \boldsymbol{\phi}))$.
6:     $t_k^{(i)}(\boldsymbol{\theta}, \boldsymbol{\phi}) \propto q_k^{(i)}(\boldsymbol{\theta}, \boldsymbol{\phi})/q_{\backslash k}^{(i-1)}(\boldsymbol{\theta}, \boldsymbol{\phi})$
7:     $q^{(i)}(\boldsymbol{\theta}, \boldsymbol{\phi}) \propto q_{\backslash k}^{(i-1)}(\boldsymbol{\theta}, \boldsymbol{\phi})t_k^{(i)}(\boldsymbol{\theta}, \boldsymbol{\phi})$
8: **end for**

## 5 PVI for Sparse Gaussian Processes

Consider a GP, $f \sim \mathcal{GP}(m_\beta, k_\beta)$, where $m_\beta$ and $k_\beta$ denote the mean and covariance functions with hyperparameters $\boldsymbol{\beta}$. We model $K$ partitions of input-output observations $\{(\mathbf{X} \in \mathcal{X}^{N_k}, \mathbf{y}_k \in \mathcal{Y}^{N_k})\}_{k=1}^{K}$ as

$$p(\mathbf{y}, f, \boldsymbol{\beta}) = p(\boldsymbol{\beta})p(f|\boldsymbol{\beta}) \prod_{k=1}^{K} \prod_{n=1}^{N_k} p(\mathbf{y}_{kn}|f(\mathbf{x}_{kn}), \boldsymbol{\beta}). \tag{8}$$

Following standard variational sparse GP approximations (Titsias, 2009; Matthews et al., 2016) and PVI, we consider an approximate posterior of the form

$$q(f, \boldsymbol{\beta}|\mathbf{Z}) \propto q(\boldsymbol{\beta}) \underbrace{p(f|\boldsymbol{\beta}) \prod_{k=1}^{K} t_k(\mathbf{u}|\mathbf{Z}, \boldsymbol{\beta})}_{q(f|\boldsymbol{\beta}, \mathbf{Z})} \tag{9}$$

where $q(\boldsymbol{\beta}) \propto p(\boldsymbol{\beta}) \prod_{k=1}^{K} t_k(\boldsymbol{\beta})$, and $\mathbf{Z} \in \mathcal{X}^M$ denotes a shared set of inducing locations with function outputs $\mathbf{u} = f(\mathbf{Z})$. Since each factor is defined over $\mathbf{u}$, we cannot locally optimise the shared inducing locations $\mathbf{Z}$ (they are equivalent to $\boldsymbol{\phi}$ in Equation (4)). A natural alternative is to define each clients' factor over the function output at a different set of inducing locations, such that $q(f|\boldsymbol{\beta}) = p(f|\boldsymbol{\beta}) \prod_{k=1}^{K} t_k(\mathbf{u}_k|\mathbf{Z}_k, \boldsymbol{\beta})$ where $\mathbf{u}_k = f(\mathbf{Z}_k)$. Whilst this addresses the problem of inducing location optimisation, we show in the appendix that this imposes a computational complexity of $\mathcal{O}((\sum_{k=1}^{K} M_k)^3)$. This is cubic[3] in $K$, and so quickly becomes prohibitive for a moderate number of partitions.

Instead, following the framework established in Section 4, we consider the HVM

$$q(f|\boldsymbol{\beta}) = \int \underbrace{p(f|\boldsymbol{\beta}) \prod_{k=1}^{K} t_k(\mathbf{u}|\mathbf{Z}, \boldsymbol{\beta})}_{q(f|\mathbf{Z}, \boldsymbol{\beta})} \underbrace{\prod_{k=1}^{K} t_k(\mathbf{Z})}_{q(\mathbf{Z})} d\mathbf{Z}. \tag{10}$$

Whilst the factors $t_k(\boldsymbol{\beta})$ and $t_k(\mathbf{Z})$ can take simple forms such as mean-field Gaussian distributions, it is important to take care when considering the dependency of $\mathbf{Z}$ and $\boldsymbol{\beta}$ in the factors $t_k(\mathbf{u}|\mathbf{Z}, \boldsymbol{\beta})$. In particular, since $\mathbf{Z}$ is being treated stochastically it is important that this does not negatively impact the quality of the approximate posterior.

---

[3]Consider $M_k = M \; \forall k$. In this case, we have $\left(\sum_{k=1}^{K} M\right)^3 = K^3 M^3$.

## 5.1 Decoupled Pseudo-Observation Parameterisation

A naïve approach would be to use the standard parameterisation $t_k(\mathbf{u}|\mathbf{Z}, \boldsymbol{\beta}) \propto \mathcal{N}(\mathbf{u}; \mathbf{m}_k, \mathbf{S}_k)$. Whilst this form is convenient to work with, thus widely used, it effectively assumes that for all values of $\mathbf{Z} \sim q(\mathbf{Z})$ the approximate likelihood is constant.[4] This assumption is inappropriate if the variance of $q(\mathbf{Z})$ is non-negligible, resulting in poor predictive performance.

Instead, consider form of the Gaussian approximate likelihood which minimises the KL-divergence $q^*(f|\mathbf{Z}) = \arg\min_{q(f|\mathbf{Z})} \mathrm{KL}\left[q(f|\mathbf{Z})||p(f|\mathbf{y})\right]$ for any inducing locations $\mathbf{Z} \in \mathcal{X}^M$:

$$t^*(\mathbf{u}|\mathbf{Z}) \propto \mathcal{N}\left(\mathbf{K_{XZ}K_{ZZ}^{-1}u}; \mathbf{m}, \mathbf{D}\right) \tag{11}$$

where $\mathbf{D}$ is a diagonal covariance matrix.[5] In the case of a Gaussian likelihood $p(y_n|f(\mathbf{x}_n)) = \mathcal{N}\left(y_n; f(\mathbf{x}_n), \sigma^2\right)$, we have $m_n = y_n$ and $D_{nn} = \sigma^2$. A derivation of this result is provided in Ashman et al. (2020). The result is analogous to that of Opper & Archambeau (2009) who derive the optimal form for non-sparse variational GP approximations, and suggests the use of $t_k(\mathbf{u}|\mathbf{Z}, \boldsymbol{\beta}) \propto \mathcal{N}\left(\mathbf{K_{X_kZ}K_{ZZ}^{-1}u}; \mathbf{m}_k, \mathbf{D}_k\right)$. This form has been used previously (Ashman et al., 2020; Jazbec et al., 2021; Adam et al., 2021), and can be thought of as optimally projecting the data onto the given inducing locations $\mathbf{Z}$ using the hyperparameters $\boldsymbol{\beta}$.

However note that Equation (11) requires use of clients' data, $\mathbf{X}$, when evaluating the approximate posterior, which both violates the restrictions of federated learning and prevents mini-batching being using during optimisation. Instead, we propose the use of an approximate likelihood which minimises the KL-divergence for any inducing locations $\mathbf{Z} \in \mathcal{X}^M$ if we pretend that $\mathbf{V}_k \in \mathcal{X}^{M_k}$ are our true input locations, $\mathbf{m}_k \in \mathbb{R}^{M_k}$ are our true observations, and $\mathbf{S}_k \in \mathbb{R}^{M_k \times M_k}$ is our model's true observation covariance, such that

$$t_k(\mathbf{u}|\mathbf{Z}, \boldsymbol{\beta}) \propto \mathcal{N}\left(\mathbf{K_{V_kZ}K_{ZZ}^{-1}u}; \mathbf{m}_k, \mathbf{S}_k\right). \tag{12}$$

We refer to the set $\{\mathbf{V}_k, \mathbf{m}_k, \mathbf{S}_k\}$ as 'pseudo-observations', and this parameterisation as the decoupled pseudo-observation (DPO) parameterisation. Unlike the naïve standard parameterisation $t_k(\mathbf{u}|\mathbf{Z}, \boldsymbol{\beta}) \propto \mathcal{N}(\mathbf{u}; \mathbf{m}_k, \mathbf{S}_k)$—which we refer to as the coupled pseudo-observation (CPO) parameterisation, as we parameterise the pseudo-observations *at* the inducing locations $\mathbf{Z}$—in the DPO parameterisation the pseudo-observations are decoupled from the inducing locations. Observe the close correspondence between Equation (12) and Equation (11): the DPO parameterisation has the same form as the optimal approximate likelihood, but replaces all private data with variational parameters (the pseudo-observations). This has three important consequences: 1. the approximate likelihood is dependent on $\mathbf{Z}$; 2. stochastic optimisation of the variational lower-bound can be performed; and 3. privacy constraints are satisfied.

## 5.2 Comparison to Bayesian FITC

As far as we are aware, the only other work that utilises probabilistic treatment of inducing locations $\mathbf{Z}$ is the Bayesian FITC method (Rossi et al., 2021). There are important differences between the two approaches. First, we construct a variational approximation to the exact GP posterior that decomposes in a manner amenable to the application of PVI, enabling federated learning to be performed. In contrast, they perform Hamiltonian Monte Carlo (HMC) (Neal et al., 2011) for inference. HMC demands the use of the entire dataset, preventing its application in the federated learning setting. Second, their approach is based on a Bayesian treatment of inducing locations in the fully-independent training conditional (FITC) approximation (Snelson & Ghahramani, 2005; Quinonero-Candela & Rasmussen, 2005), which, dissimilar to our approach, approximates the GP model rather than its posterior (Bui et al., 2017). The distribution over inducing locations that they obtain arises due to the treatment of inducing locations as model parameters, rather than reformulation as auxiliary variables in a HVM.

---

[4]This can be thought of as assuming the same observations for any set of inputs $\mathbf{Z} \sim q(\mathbf{Z})$.

[5]Note that this can also be written in the form $t^*(\mathbf{u}|\mathbf{Z}) \propto \prod_{n=1}^N \mathcal{N}\left(\mathbf{k_{x_nZ}K_{ZZ}^{-1}u}; m_n, D_{nn}\right)$, which is more commonly seen in existing literature.

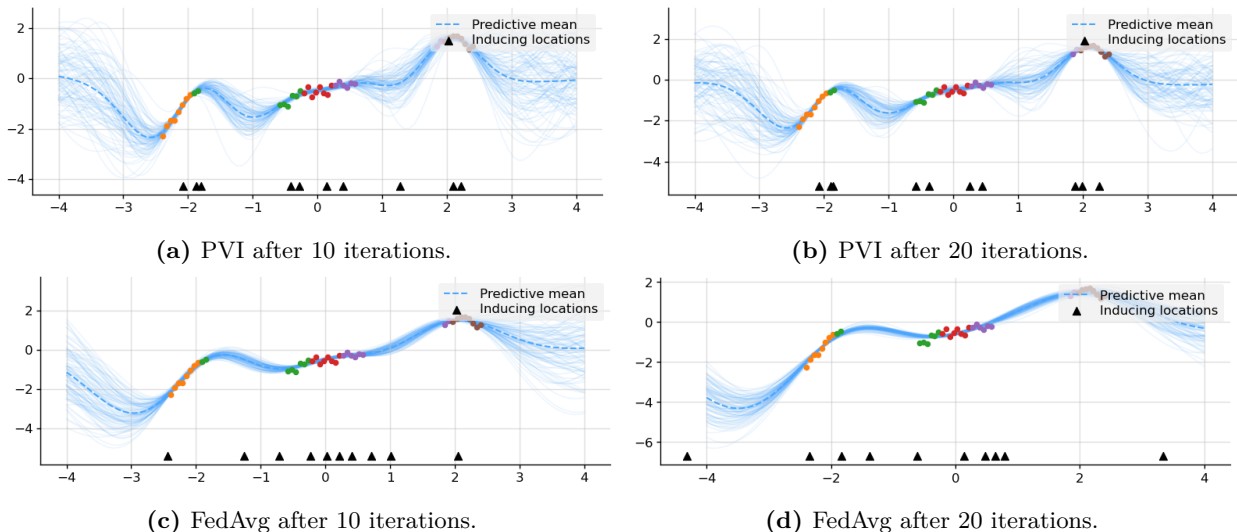

**Figure 1:** Samples from the SGP posterior predictive distribution when performing federated learning of inducing locations using FedAvg (bottom) and our PVI approach (top). The partitions of data are shown in different colours, and the learnt inducing locations ($\mathbb{E}_{q(\mathbf{Z})}[\mathbf{Z}]$ in the case of the PVI) are shown as black triangles along the bottom of each figure.

## 6 Experiments

For all experiments, we employ a squared exponential (SE) kernel with a separate lengthscale for each input dimension. The prior distribution over hyperparameters (kernel lengthscale, kernel scale and observation noise) are set to be log-normal distributions with zero mean and unit variance, and log-normal variational approximations are employed. Predictive performances are evaluated using the Monte-Carlo estimate $q(y_*) \approx \frac{1}{S} \sum_{s=1}^{S} \int p(y_* | f(\mathbf{x}_*)) q(f(\mathbf{x}_*) | \boldsymbol{\beta}_s, \mathbf{Z}_s) df(\mathbf{x}_*)$, where $\boldsymbol{\beta}_s, \mathbf{Z}_s \sim q(\boldsymbol{\beta}, \mathbf{Z})$. We use $S = 100$ samples.

### 6.1 Synthetic Regression Data

In this experiment, we seek to demonstrate that using FedAvg to learn inducing locations is ineffective. We construct a synthetic regression dataset partitioned into five smaller subsets, as illustrated in Figure 1. We perform FedAvg optimisation of the inducing locations as described in Algorithm 1, choosing $J = 1000$ inner optimisation iterations. Figures 1c and 1d compares the learnt approximate posterior distribution after 10 and 20 communications. We compare this to the the learnt approximate posterior when using PVI to learn inducing locations (described in Section 5 and Algorithm 2), shown in Figures 1a and 1d. Both methods use the DPO parameterisation and hyperparameters are fixed to their true values.

We see that even after 10 iterations of sequential PVI the inducing locations learnt when using FedAvg begin to exhibit undesirable behaviour by moving away from the data, resulting in poor uncertainty estimates in-between partitions of data. This effect is exacerbated after 20 iterations. In Figure 2, we compare the effect the number of FedAvg iterations ($J$) has on the training log-likelihood during federated learning. Observe that only for very small $J$ does the log-likelihood converge, resulting in poor communication efficiency as many rounds of local optimisation are needed. This contrasts with the very fast convergence when using PVI to learn inducing locations. This undesirable behaviour of FedAvg can be attributed to the significant difference between $\mathbf{Z}_{k,J}^{(i)}$ and the updated $\mathbf{Z}^{(i)}$ at each round of local-optimisation: 1. there is no guarantee that the weighted average $\mathbf{Z}^{(i)} = \frac{N - N_k}{N} \mathbf{Z}^{(i-1)} + \frac{N_k}{N} \mathbf{Z}_{k,J}^{(i)}$ models the data well; and 2. the variational parameters of $t_k^{(i)}(\mathbf{u}|\mathbf{Z})$ are learnt assuming $\mathbf{Z} = \mathbf{Z}_{k,J}^{(i)}$, whereas they are deployed at test time using $\mathbf{Z} = \mathbf{Z}^{(i)}$.

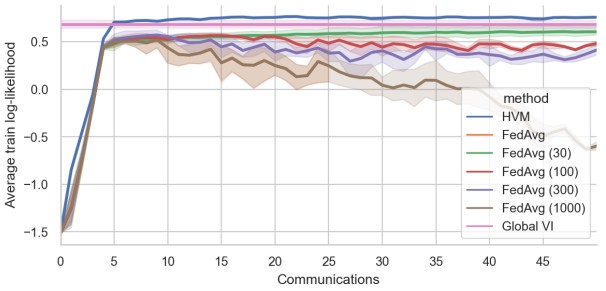 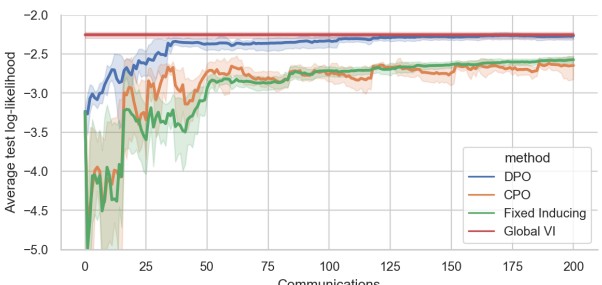

**Figure 2:** Average train log-likelihood for the synthetic regression dataset during federated learning. The error bars show the standard error over five random seeds.

**Figure 3:** Average test log-likelihood for the European weather regression dataset during federated learning. The error bars show the standard error over the give random seeds.

## 6.2 Real-World Regression Data

In this section, we evaluate the effectiveness of PVI for federated learning of sparse GPs on a number of real-world regression tasks. For each experiment, we compare the performance of the DPO parameterisation with diagonal pseudo-noise to the coupled pseudo-observation (CPO) parameterisation, where $t_k(\mathbf{u}|\mathbf{Z}, \boldsymbol{\beta}) = \mathcal{N}(\mathbf{u}; \mathbf{m}_k, \mathbf{S}_k)$, and to the performance of sparse GPs when the inducing points are fixed at their random initialisation. For DPO and CPO we choose $\tilde{p}(\mathbf{Z}) = \mathcal{N}(\mathbf{Z}; \mathbf{0}, \mathbf{I})$ and restrict each factor $t_k(\mathbf{Z})$ to be a fully-factorised Gaussian. We construct the local free-energies using $\alpha = 0.1$ in all experiments. Throughout, we provide results for sparse GPs optimised using the entire dataset to serve as a gold-standard. The number of pseudo-observations in the DPO parameterisation, $M_k$, is chosen such that $M_k < N_k$. Although this restricts the approximate posterior in the sense that we cannot recover the optimal sparse GP approximation given in Equation (11), it ensures the pseudo-observations summarise the data rather than simply learning it. Provided $M_k$ was significantly larger than $M$, we did not observe noticeable performance improvements through increasing $M_k$. We provide complete experimental details in Appendix D.

### 6.2.1 UCI Regression

Figure 4 compares the performance of PVI applied to sparse GPs on eight UCI regression datasets, each of which are partitioned into $K = 10$ homogeneous splits. We use $M = 100$ inducing locations and $M_k = \min(0.8|\mathcal{D}_k|, 500)$ pseudo-observations for the DPO parameterisation. As we do not have access to the training data, we select fixed inducing locations at random by drawing samples from a standard normal. The predictive performance is evaluated on 10 different splits from (Hernández-Lobato & Adams, 2015; Gal & Ghahramani, 2016) and we plot the mean and standard errors after 100 communications.

The use of the DPO parameterisation outperforms the use of the CPO parameterisation on all datasets, and is comparable to a sparse GP learnt using standard VI in the centralised setting.[6] This i) demonstrates the effectiveness of the DPO parameterisation; and ii) indicates that worsening the variational bound does not significantly harm the quality of the learnt approximate posterior. Fixing inducing locations performs poorly on all datasets, owing largely to the poor choice of initialisation. However, sensible initialisations of inducing locations often rely on access to the training data,[7] which is not available in the federated learning setting as it typically forbids datapoints to be communicated from clients to the server. Hence, this is unavoidable.

### 6.2.2 European Weather Regression

We construct a dataset of minimum temperature recordings on 1st January 1980 taken by 1648 weather stations situated across Europe.[8] Inhomogeneous partitions are constructed by grouping together data

---

[6]We also note that these results are comparable to those reported elsewhere (e.g. see (Salimbeni & Deisenroth, 2017; Bui et al., 2016; Hernández-Lobato & Adams, 2015)).

[7]For example, k-means clustering.

[8]These data are freely available at www.ncei.noaa.gov.

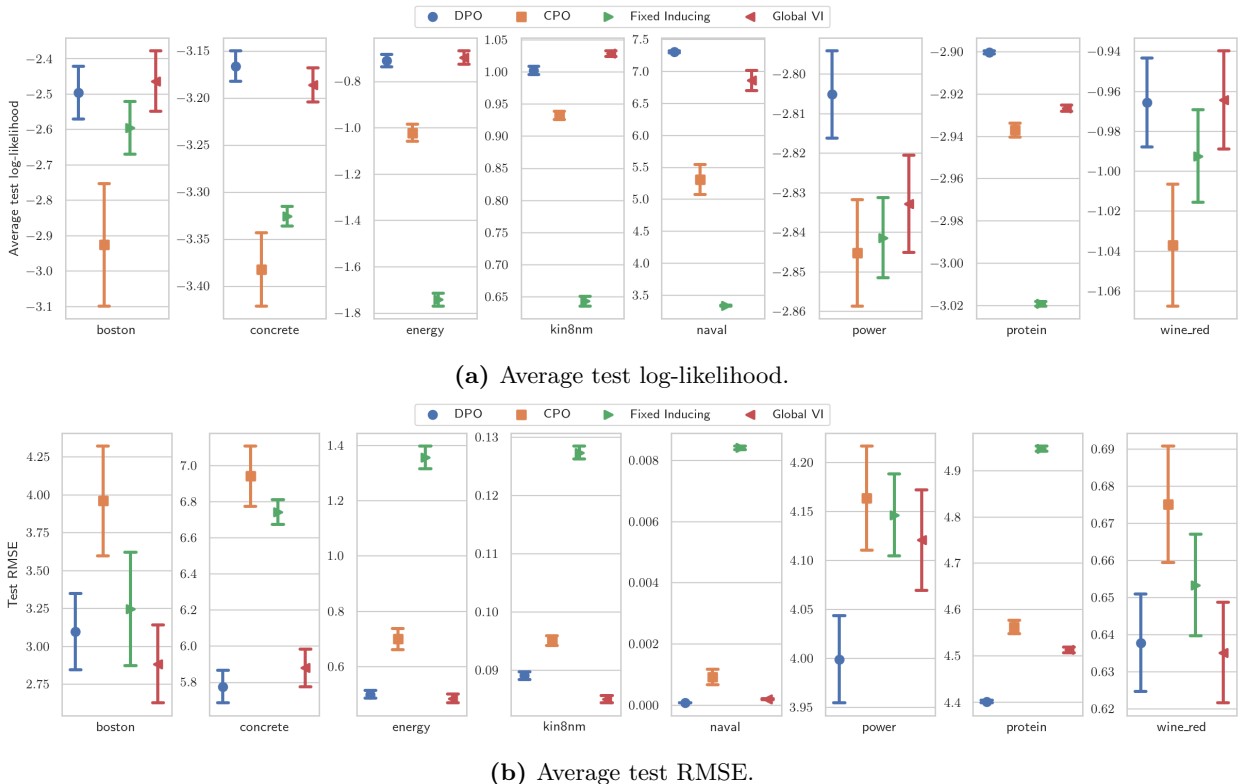

**(a)** Average test log-likelihood.

**(b)** Average test RMSE.

**Figure 4:** Performance metrics for each method on eight of the UCI regression datasets. The error bars show the standard error over the 10 different splits.

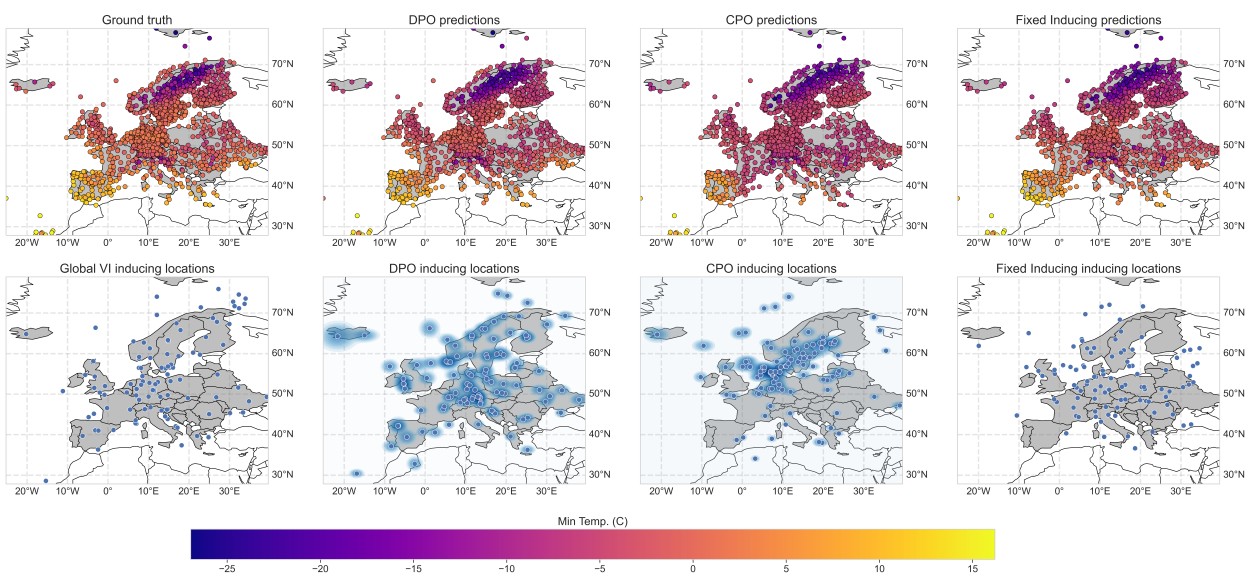

**Figure 5:** Top: predictive means on the European weather station dataset. Bottom: learnt inducing locations of each method. For methods involving a distribution, we plot $\sum_{m=1}^{M} \log q(\mathbf{z}_m)$. The data are partitioned according to the country of each weather station.

within each country, leading to 32 partitions in total. The dataset and partitioning is illustrated in Figure 5. We model the data using a sparse GP with $M = 100$ inducing locations and $M_k = \min(0.8|\mathcal{D}_k|, 100)$ pseudo-observations for the DPO parameterisation. The fixed inducing locations are drawn by uniformly sampling within the input domain and rejecting points that lie outside of country borders. In Figure 3 we plot the training dynamics of each method, showing the average test log-likelihood and standard errors across five different seeds. Figure 5 compares the predictions and inducing locations of each method after 200 communications with the ground truth temperature values.

Similar to the UCI regression tasks, we observe that with the use of the DPO parameterisation the federated sparse GP converges towards the performance of a sparse GP trained using global VI, whereas using the CPO parameterisation and fixed inducing points results in significantly worse performance. Moreover, we see that the DPO-learnt $q(\mathbf{Z})$ resembles the distribution of inducing locations learnt by global VI, with the addition of some stochasticity. The CPO-learnt $q(\mathbf{z})$ does not, and is instead more densely clustered around areas with a high density of observations resulting in poor predictive performance elsewhere.

## 7 Conclusion

In this work, we developed an effective federated learning method for sparse Gaussian processes (sparse GPs) based on partitioned variational inference (PVI). Application of PVI to sparse GPs is non-trivial since inducing locations are shared across client-specific factors. By instead viewing inducing locations as auxiliary variables in a hierarchical variational model, a distribution over inducing locations can be maintained to which PVI can be applied to in the standard way. We develop a novel form for the sparse GP variational approximation which we refer to as the decoupled pseudo-observation parameterisation (DPO). The DPO parameterisation is defined as the optimal Gaussian variational approximation given some 'pseudo-observations' and inducing locations. Importantly, it incorporates the dependency between the approximate posterior and choice of inducing locations, which improves both the communication-efficiency of federated learning and the quality of the approximate posterior when inducing locations are stochastic. We demonstrate the efficacy of our approach to federated learning of sparse GPs on a number of experiments involving real-world data.

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

## A    Defining the Auxiliary Likelihood

In the seminal work of Ranganath et al. (2016), HVMs are applied to latent variable models for which $\boldsymbol{\theta}$ is decomposed into per-datapoint latent variables which are relatively low-dimensional. Under such conditions, we can construct $r(\boldsymbol{\phi}|\boldsymbol{\theta})$ using an inference network to parameterise the base distribution of a normalising flow. Incorporating the dependency of $\boldsymbol{\theta}$ in the auxiliary likelihood is essential for the effectiveness of hierarchical models in VI. Without any dependency on $\boldsymbol{\theta}$, the optimum is achieved at $r(\boldsymbol{\phi}) = q(\boldsymbol{\phi}) = \delta(\boldsymbol{\phi} - \boldsymbol{\phi}^*)$, i.e. the solution obtained with no hierarchical model. This behaviour manifests itself in PVI through the collapse of each $t_k(\boldsymbol{\phi})$ to a delta function, leading to degenerate behaviour when aggregated.

Unfortunately, we are interested in applying PVI for variational approximations in which there is no clear method for incorporating the dependency of $\boldsymbol{\theta}$ on $\boldsymbol{\phi}$ in the auxiliary likelihood. However, consider the construction $q(\boldsymbol{\phi}) = \tilde{p}(\boldsymbol{\phi}) \prod_{k=1}^{K} t_k(\boldsymbol{\phi})$, where $\tilde{p}(\boldsymbol{\phi})$ is some arbitrary distribution. Using $r(\boldsymbol{\phi}|\boldsymbol{\theta}) = \tilde{p}(\boldsymbol{\phi})$ gives

$$\tilde{\mathcal{L}} = \mathbb{E}_q \left[ \log p(\mathbf{y}|\boldsymbol{\theta}) - \mathrm{KL}\left[q(\boldsymbol{\theta}|\boldsymbol{\phi})||p(\boldsymbol{\theta})\right] \right] - \mathrm{KL}\left[q(\boldsymbol{\phi})||\tilde{p}(\boldsymbol{\phi})\right]. \tag{13}$$

Although we have worsened the standard variational bound, and have therefore sacrificed the accuracy of the approximate posterior, the use of this sub-optimal solution avoids the degenerate behaviour of $q(\boldsymbol{\phi})$—and therefore each $t_k(\boldsymbol{\phi})$—collapsing to a delta.

## B    Relationship Between PVI and Global VI Fixed Points

Let $\boldsymbol{\eta}_q$ be the variational parameters of $q(\boldsymbol{\theta}, \boldsymbol{\phi})$ and $\boldsymbol{\eta}_q^*$ be the solution to the global VI optimisation problem:

$$\boldsymbol{\eta}_q^* = \arg\max_{\boldsymbol{\eta}_q} \underbrace{\mathbb{E}_q \left[ \log p(\mathbf{y}|\boldsymbol{\theta}) - \mathrm{KL}\left[q(\boldsymbol{\theta}|\boldsymbol{\phi})||p(\boldsymbol{\theta})\right] \right] - \alpha\mathrm{KL}\left[q(\boldsymbol{\phi})||\tilde{p}(\boldsymbol{\phi})\right]}_{\tilde{\mathcal{L}}_\alpha} \tag{14}$$

and the variational parameters of $q^*(\boldsymbol{\theta}, \boldsymbol{\phi})$. Define $q_\alpha(\boldsymbol{\theta}, \boldsymbol{\phi}) = \frac{1}{\mathcal{Z}_{q_\alpha(\boldsymbol{\theta}, \boldsymbol{\phi})}} q(\boldsymbol{\theta}|\boldsymbol{\phi}) q(\boldsymbol{\phi})^\alpha$ and $q_{\alpha, \backslash k}(\boldsymbol{\theta}, \boldsymbol{\phi}) = \frac{1}{\mathcal{Z}_{q_{\alpha, \backslash k}(\boldsymbol{\theta}, \boldsymbol{\phi})}} p(\boldsymbol{\theta}) \tilde{p}(\boldsymbol{\phi})^\alpha \prod_{m \neq k} t_m(\boldsymbol{\theta}|\boldsymbol{\phi}) t_m(\boldsymbol{\phi})^\alpha$.

$$\frac{d\mathcal{L}_k}{d\boldsymbol{\eta}_q} = \frac{d}{d\boldsymbol{\eta}_q} \int q(\boldsymbol{\theta}, \boldsymbol{\phi}) \log \frac{\mathcal{Z}_{q^*_{\alpha,\backslash k}(\boldsymbol{\theta},\boldsymbol{\phi})} q^*(\boldsymbol{\theta}, \boldsymbol{\phi}) p(\mathbf{y}_k|\boldsymbol{\theta})}{q(\boldsymbol{\theta}|\boldsymbol{\phi}) q(\boldsymbol{\phi})^\alpha t^*_k(\boldsymbol{\theta}|\boldsymbol{\phi}) t^*(\boldsymbol{\phi})^\alpha} d\boldsymbol{\theta} d\boldsymbol{\phi}$$

$$= \frac{d}{d\boldsymbol{\eta}_q} \int q(\boldsymbol{\theta}, \boldsymbol{\phi}) \log \frac{p(\mathbf{y}_k|\boldsymbol{\theta})}{t^*_k(\boldsymbol{\theta}|\boldsymbol{\phi}) t^*_k(\boldsymbol{\phi})^\alpha} d\boldsymbol{\theta} d\boldsymbol{\phi} + \frac{d}{d\boldsymbol{\eta}_q} \int q(\boldsymbol{\phi}) \int q(\boldsymbol{\theta}, \boldsymbol{\phi}) \log \frac{q^*(\boldsymbol{\theta}|\boldsymbol{\phi})}{q(\boldsymbol{\theta}|\boldsymbol{\phi})} d\boldsymbol{\theta} d\boldsymbol{\phi}$$

$$+ \frac{d}{d\boldsymbol{\eta}_q} \int q(\boldsymbol{\phi}) \log q^*(\boldsymbol{\phi}) d\boldsymbol{\phi} - \alpha \frac{d}{d\boldsymbol{\eta}_q} \int q(\boldsymbol{\phi}) \log q(\boldsymbol{\phi}) d\boldsymbol{\phi} + \underbrace{\frac{d}{d\boldsymbol{\eta}_q} \log \mathcal{Z}_{q^*_{\alpha,\backslash k}(\boldsymbol{\theta},\boldsymbol{\phi})}}_{0}$$

$$= \frac{d}{d\boldsymbol{\eta}_q} \int q(\boldsymbol{\theta}, \boldsymbol{\phi}) \log \frac{p(\mathbf{y}_k|\boldsymbol{\theta})}{t^*_k(\boldsymbol{\theta}|\boldsymbol{\phi}) t^*_k(\boldsymbol{\phi})^\alpha} d\boldsymbol{\theta} d\boldsymbol{\phi} + \int \frac{dq(\boldsymbol{\phi})}{d\boldsymbol{\eta}_q} \int q(\boldsymbol{\theta}|\boldsymbol{\phi}) \log \frac{q^*(\boldsymbol{\theta}|\boldsymbol{\phi})}{q(\boldsymbol{\theta}|\boldsymbol{\phi})} d\boldsymbol{\theta} d\boldsymbol{\phi}$$

$$+ \int q(\boldsymbol{\phi}) \int \frac{dq(\boldsymbol{\theta}|\boldsymbol{\phi})}{d\boldsymbol{\eta}_q} \log \frac{q^*(\boldsymbol{\theta}|\boldsymbol{\phi})}{q(\boldsymbol{\theta}|\boldsymbol{\phi})} d\boldsymbol{\theta} d\boldsymbol{\phi} - \int q(\boldsymbol{\phi}) \int \underbrace{\frac{dq(\boldsymbol{\theta}|\boldsymbol{\phi})}{d\boldsymbol{\eta}_q}}_{0} d\boldsymbol{\theta} d\boldsymbol{\phi}$$

$$+ \int \frac{dq(\boldsymbol{\phi})}{d\boldsymbol{\eta}_q} \log q^*(\boldsymbol{\phi}) d\boldsymbol{\phi} - \alpha \int \frac{dq(\boldsymbol{\phi})}{d\boldsymbol{\eta}_q} \log q(\boldsymbol{\phi}) d\boldsymbol{\phi} - \alpha \int \underbrace{\frac{dq(\boldsymbol{\phi})}{d\boldsymbol{\eta}_q}}_{0} d\boldsymbol{\phi}$$

$$= \frac{d}{d\boldsymbol{\eta}_q} \int q(\boldsymbol{\theta}, \boldsymbol{\phi}) \log \frac{p(\mathbf{y}_k|\boldsymbol{\theta})}{t^*_k(\boldsymbol{\theta}|\boldsymbol{\phi}) t^*_k(\boldsymbol{\phi})^\alpha} d\boldsymbol{\theta} d\boldsymbol{\phi} + \int \frac{dq(\boldsymbol{\theta}, \boldsymbol{\phi})}{d\boldsymbol{\eta}_q} \int q(\boldsymbol{\theta}|\boldsymbol{\phi}) \log \frac{q^*(\boldsymbol{\theta}|\boldsymbol{\phi})}{q(\boldsymbol{\theta}|\boldsymbol{\phi})} d\boldsymbol{\theta} d\boldsymbol{\phi}$$

$$+ \int \frac{dq(\boldsymbol{\phi})}{d\boldsymbol{\eta}_q} \log \frac{q^*(\boldsymbol{\theta})}{q(\boldsymbol{\phi})^\alpha}. \tag{15}$$

Thus, at convergence when $\boldsymbol{\eta}_q = \boldsymbol{\eta}^*_q$,

$$\left. \frac{d\mathcal{L}_k}{d\boldsymbol{\eta}_q} \right|_{\boldsymbol{\eta}_q=\boldsymbol{\eta}^*_q} = \left. \frac{d}{d\boldsymbol{\eta}_q} \int q(\boldsymbol{\theta}, \boldsymbol{\phi}) \log \frac{p(\mathbf{y}_k|\boldsymbol{\theta})}{t^*_k(\boldsymbol{\theta}|\boldsymbol{\phi}) t^*_k(\boldsymbol{\phi})^\alpha} d\boldsymbol{\theta} d\boldsymbol{\phi} \right|_{\boldsymbol{\eta}_q=\boldsymbol{\eta}^*_q}$$

$$+ (1-\alpha) \left. \frac{d}{d\boldsymbol{\eta}_q} \int q(\boldsymbol{\phi}) \log q^*(\boldsymbol{\phi}) d\boldsymbol{\phi} \right|_{\boldsymbol{\eta}_q=\boldsymbol{\eta}^*_q} \tag{16}$$

Summing both sides over $k$ gives

$$\left. \sum_{k=1}^K \frac{d\mathcal{L}_k}{d\boldsymbol{\eta}_q} \right|_{\boldsymbol{\eta}_q=\boldsymbol{\eta}^*_q} = \left. \frac{d}{d\boldsymbol{\eta}_q} \int q(\boldsymbol{\theta}, \boldsymbol{\phi}) \log \frac{\prod_{k=1}^K p(\mathbf{y}_k|\boldsymbol{\theta})}{\prod_{k=1}^K t^*_k(\boldsymbol{\theta}|\boldsymbol{\phi}) t^*_k(\boldsymbol{\phi})^\alpha} d\boldsymbol{\theta} d\boldsymbol{\phi} \right|_{\boldsymbol{\eta}_q=\boldsymbol{\eta}^*_q}$$

$$+ K(1-\alpha) \left. \frac{d}{d\boldsymbol{\eta}_q} \int q(\boldsymbol{\phi}) \log q^*(\boldsymbol{\phi}) d\boldsymbol{\phi} \right|_{\boldsymbol{\eta}_q=\boldsymbol{\eta}^*_q}$$

$$= \left. \frac{d}{d\boldsymbol{\eta}_q} \int q(\boldsymbol{\theta}, \boldsymbol{\phi}) \log \frac{p(\boldsymbol{\theta}) \tilde{p}(\boldsymbol{\phi})^\alpha \prod_{k=1}^K p(\mathbf{y}_k|\boldsymbol{\theta})}{q^*_\alpha(\boldsymbol{\theta}, \boldsymbol{\phi})} d\boldsymbol{\theta} d\boldsymbol{\phi} \right|_{\boldsymbol{\eta}_q=\boldsymbol{\eta}^*_q} \tag{17}$$

$$- \underbrace{\frac{d}{d\boldsymbol{\eta}_q} \log \mathcal{Z}_{q^*_\alpha(\boldsymbol{\theta},\boldsymbol{\phi})}}_{0} + K(1-\alpha) \left. \frac{d}{d\boldsymbol{\eta}_q} \int q(\boldsymbol{\phi}) \log q^*(\boldsymbol{\phi}) d\boldsymbol{\phi} \right|_{\boldsymbol{\eta}_q=\boldsymbol{\eta}^*_q}$$

where we have defined $q^*_\alpha(\boldsymbol{\theta}, \boldsymbol{\phi}) = \frac{1}{\mathcal{Z}_{q^*_\alpha(\boldsymbol{\theta},\boldsymbol{\phi})}} p(\boldsymbol{\theta}) \tilde{p}(\boldsymbol{\phi})^\alpha \prod_{k=1}^K t_k(\boldsymbol{\theta}|\boldsymbol{\phi}) t_k(\boldsymbol{\phi})^\alpha$.

Now, consider the derivative of the global variational lower-bound $\tilde{\mathcal{L}}_k$:

$$\frac{d\tilde{\mathcal{L}}}{d\boldsymbol{\eta}_q} = \frac{d}{d\boldsymbol{\eta}_q} \int q(\boldsymbol{\theta}, \boldsymbol{\phi}) \log \frac{p(\boldsymbol{\theta})\tilde{p}(\boldsymbol{\phi})^\alpha \prod_{k=1}^K p(\mathbf{y}_k|\boldsymbol{\theta})}{q_\alpha(\boldsymbol{\theta}, \boldsymbol{\phi})} d\boldsymbol{\theta} d\boldsymbol{\phi}$$

$$= \int \frac{dq(\boldsymbol{\theta}, \boldsymbol{\phi})}{d\boldsymbol{\eta}_q} \log \frac{p(\boldsymbol{\theta})\tilde{p}(\boldsymbol{\phi})^\alpha \prod_{k=1}^K p(\mathbf{y}_k|\boldsymbol{\theta})}{q_\alpha(\boldsymbol{\theta}, \boldsymbol{\phi})} d\boldsymbol{\theta} d\boldsymbol{\phi} - \int q(\boldsymbol{\phi}) \int \cancel{\frac{dq(\boldsymbol{\theta}|\boldsymbol{\phi})}{d\boldsymbol{\eta}_q}}^{0} d\boldsymbol{\theta} d\boldsymbol{\phi} \tag{18}$$

$$- \alpha \int \cancel{\frac{dq(\boldsymbol{\phi})}{d\boldsymbol{\eta}_q}}^{0} d\boldsymbol{\phi}.$$

At $\boldsymbol{\eta}_q = \boldsymbol{\eta}_q^*$ we have

$$\left.\frac{d\tilde{\mathcal{L}}}{d\boldsymbol{\eta}_q}\right|_{\boldsymbol{\eta}_q=\boldsymbol{\eta}_q^*} = \int q(\boldsymbol{\theta}, \boldsymbol{\phi}) \log \frac{p(\boldsymbol{\theta})\tilde{p}(\boldsymbol{\phi})^\alpha \prod_{k=1}^K p(\mathbf{y}_k|\boldsymbol{\theta})}{q_\alpha^*(\boldsymbol{\theta}, \boldsymbol{\phi})} d\boldsymbol{\theta} d\boldsymbol{\phi} \tag{19}$$

giving

$$\sum_{k=1}^K \left.\frac{d\mathcal{L}_k}{d\boldsymbol{\eta}_q}\right|_{\boldsymbol{\eta}_q=\boldsymbol{\eta}_q^*} - \left.\frac{d\tilde{\mathcal{L}}}{d\boldsymbol{\eta}_q}\right|_{\boldsymbol{\eta}_q=\boldsymbol{\eta}_q^*} = K(1-\alpha)\left.\frac{d}{d\boldsymbol{\eta}_q} \int q(\boldsymbol{\phi}) \log q^*(\boldsymbol{\phi}) d\boldsymbol{\phi}\right|_{\boldsymbol{\eta}_q=\boldsymbol{\eta}_q^*}. \tag{20}$$

Thus, only at $\alpha = 1$ does $\sum_{k=1}^K \left.\frac{d\mathcal{L}_k}{d\boldsymbol{\eta}_q}\right|_{\boldsymbol{\eta}_q=\boldsymbol{\eta}_q^*} = \left.\frac{d\tilde{\mathcal{L}}}{d\boldsymbol{\eta}_q}\right|_{\boldsymbol{\eta}_q=\boldsymbol{\eta}_q^*} = 0$.

## C    Sparse GP Computational Complexity

A possible choice of approximate posterior for GPs in the federated learning setting is

$$q(f|\mathbf{Z}) = p(f|\mathbf{Z}) \prod_{k=1}^K t_k(\mathbf{u}_k|\mathbf{Z}_k) \tag{21}$$

where $\mathbf{u}_k = f(\mathbf{Z}_k)$ and $t_k(\mathbf{u}_k|\mathbf{Z}_k) \propto \mathcal{N}(\mathbf{u}_k; \mathbf{m}_k, \mathbf{S}_k)$. The computational complexity associated with learning and inference is dominated by computation of $q(\mathbf{u}) = \mathcal{N}(\mathbf{u}; \mathbf{m}, \boldsymbol{\Lambda}^{-1}) \propto p(\mathbf{u}|\mathbf{Z}) \prod_{k=1}^K t_k(\mathbf{u}_k|\mathbf{Z}_k)$, where $\mathbf{u} = f(\mathbf{Z}_1, \ldots, \mathbf{Z}_K)$. We can write each $t_k(\mathbf{u}_k)$ as a normal distribution over $\mathbf{u}$ with mean $\tilde{\mathbf{m}}_k \in \mathbb{R}^{\sum_{k=1}^K M_k}$ and precision $\tilde{\boldsymbol{\Lambda}}_k \in \mathbb{R}^{\sum_{k=1}^K M_k \times \sum_{k=1}^K M_k}$, where $\tilde{\mathbf{m}}_k$ has $M_k$ non-zero elements corresponding to the indices of $\mathbf{u}_k$ and $\tilde{\boldsymbol{\Lambda}}_k$ has $M_k \times M_k$ non-zero elements corresponding to block of indices corresponding to $\mathbf{u}_k$. Thus, the overall mean and precision of $q(\mathbf{u})$ is given by

$$\mathbf{m} = \boldsymbol{\Lambda}^{-1} \sum_{k=1}^K \tilde{\boldsymbol{\Lambda}}_k \tilde{\mathbf{m}}_k \quad \text{and} \quad \boldsymbol{\Lambda} = \mathbf{K}_{\mathbf{ZZ}}^{-1} + \sum_{k=1}^K \tilde{\boldsymbol{\Lambda}}_k. \tag{22}$$

This requires inverting a $\sum_{k=1}^K M_k \times \sum_{k=1}^K M_k$, which has an associated computational complexity of $\mathcal{O}\left(\left(\sum_{k=1}^K M_k\right)^3\right)$.

Instead, consider use of the DPO parameterisation

$$t_k(\mathbf{u}|\mathbf{Z}, \boldsymbol{\beta}) \propto \mathcal{N}\left(\mathbf{K}_{\mathbf{V}_k\mathbf{Z}}\mathbf{K}_{\mathbf{ZZ}}^{-1}\mathbf{u}; \mathbf{m}_k, \mathbf{S}_k\right). \tag{23}$$

We can equivalently express this as

$$t_k(\mathbf{u}|\mathbf{Z}, \boldsymbol{\beta}) \propto \mathcal{N}\left(\mathbf{u}; \tilde{\mathbf{m}}_k, \tilde{\boldsymbol{\Lambda}}_k^{-1}\right) \tag{24}$$

where

$$\tilde{\mathbf{m}}_k = \tilde{\boldsymbol{\Lambda}}_k^{-1}\mathbf{K}_{\mathbf{ZZ}}^{-1}\mathbf{K}_{\mathbf{V}_k\mathbf{Z}}\mathbf{S}_k^{-1}\mathbf{m}_k \quad \text{and} \quad \tilde{\boldsymbol{\Lambda}}_k = \mathbf{K}_{\mathbf{ZZ}}^{-1}\mathbf{K}_{\mathbf{ZV}_k}\mathbf{S}_k^{-1}\mathbf{K}_{\mathbf{V}_k\mathbf{Z}}\mathbf{K}_{\mathbf{ZZ}}^{-1}. \tag{25}$$

Thus, the computational complexity of evaluating $q(\mathbf{u})$ when using the DPO parameterisation is $\mathcal{O}\left(\sum_{k=1}^K \left(M_k^3 + M_k M^2\right) + M^3\right)$ if $\mathbf{S}_k$ is full-rank, and $\mathcal{O}\left(\sum_{k=1}^K \left(M_k M^2\right) + M^3\right)$ if $\mathbf{S}_k$ is diagonal.

## D   Experimental Details

All experiments employ sequential PVI with no damping. We perform local optimisation using the Adam optimiser (Kingma & Ba, 2014) with a learning rate of 1e-2 and $\beta = (0.9, 0.999)$ until convergence of the local free-energy. For variational parameters, we initialise all precisions to be 100 and all means to be zero. All datasets are standardised. We describe the datasets used in more detail below.

**UCI Regression**   Table 1 details the properties of the UCI regression datasets.

| Dataset | $N$ | $D$ |
|---:|---:|---:|
| boston | 506 | 13 |
| concrete | 1030 | 8 |
| energy | 768 | 8 |
| kin8nm | 8192 | 8 |
| naval | 11934 | 16 |
| power | 9568 | 4 |
| protein | 9568 | 4 |
| wine red | 1588 | 11 |
| yacht | 308 | 6 |

**Table 1:** Properties of the UCI regression datasets. $N$ denotes the number of datapoints and $D$ denotes the number of input dimensions.

**European Weather Station**   The input domain is 3-dimensional, consisting of longitude, latitude and elevation. The countries included in the dataset are: Albania, Austria, Belgium, Bosnia and Herzegovina, Belarus, Bulgaria, Denmark, Ireland, Czech Republic, Finland, France, Germany, Greece, Croatia, Hungary, Iceland, Italy, Latvia, Lithuania, Slovakia, Luxembourg, Moldova, Malta, Netherlands, Norway, Poland, Portugal, Serbia, Romania, Slovenia, Spain, Sweden, Switzerland, United Kingdom and Ukraine.

