# OpenReview forum: "Federated Learning of Sparse Gaussian Processes"
_TMLR — Rejected by TMLR_

### Review · Reviewer_wWdG · 2024-01-15

**Summary Of Contributions:**

This paper presents a probabilistic approach to federated learning using partitioned variational inference for sparse Gaussian processes (SGP).

More specifically, the paper argued that the inducing inputs that parameterize an SGP are often treated as variational parameters, which needs to be shared across clients but the original PVI mechanism would only allow model parameters (e.g., mean & kernel parameters) to be learned in a federated manner, leaving the other set of variational parameters to be completely fitted to local data & forgetting contributions for others.

To fix this, the paper proposes to treat the inducing inputs as random variables instead, which can be marginalized out. This inspires a new parameterization for sparse Gaussian processes that decouple the mean & covariance of the approximate GP posterior from the inducing inputs. This is achieved via using a previous hierarchical variational models (HVMs).

**Audience:**

Yes

**Claims And Evidence:**

No

**Requested Changes:**

Overall, the paper explores a relatively interesting approach to federated learning. But, it does not quite have a clear motivation (what values does applying sparse GP to FL provide?) & sufficient empirical evaluation (i.e., evaluation on large-scale dataset & comparison with existing SOTA in federated learning).

In addition, there are concerns in the Weakness section that need clarifications.

**Strengths And Weaknesses:**

STRENGTHS

The paper tackles the probabilistic federated learning problem with sparse Gaussian processes, which is relatively new given that much of the federated learning literature has focused primarily on neural network. This is an interesting angle, with pretty interesting mathematical idea. Overall, the idea of modeling the inducing inputs as random variables is quite intriguing & elegant to me but I have multiple reservations regarding its motivation & significance & empirical studies (see in next section)

WEAKNESSES

The paper argues that making inducing inputs shared parameters is necessary to avoid forgetting contributions from others. Is that something observable in the experiment?

Previously, it has been shown that prediction at each test input can often be improved by restricting its condition to its local neighborhood of the training data -- see http://proceedings.mlr.press/v2/snelson07a/snelson07a.pdf -- which suggests that localizing the inducing inputs might also help.

Thus, I tend to think that given global or federated model parameters, local client should be allowed to customize their inducing locations. PVI naturally allows this, why taking this away?

Another point that confuses me is: We can always rewrite sparse Gaussian processes as Gaussian processes with degenerated kernel parameterized by the inducing inputs. Doesn't it allow the inducing input to be treated as model parameters?

Otherwise, from the pure GP-wise contribution, while I agree that this HVM-based parameterization is somewhat new but its motivation is still somewhat lacking: if the ultimate goal here is to decouple the inducing inputs from the mean & covariance parameters, there has already been several solutions, e.g.

Variational Inference for Mahalanobis Distance Metrics in Gaussian Process Regression (NIPS-13)

The above work was not presented in the federated learning context (it obviously predates FL) but clearly, its formulation does not require centralizing data as model parameters are updated in stochastic batches of data. As such, I believe it is applicable to FL as is -- perhaps this is a potential baseline to compare with?

Overall, I find the position of this paper is not very well motivated. Bringing Gaussian process to federated learning for the sake of doing so is probably algorithmically interesting but beyond that, what practical value does it add to federated learning in general?

Will it perform on par with state-of-the-art, non-GP FL method on the other large-scale benchmarks?

---

> ### Author Response · Authors · 2024-03-13
> **Response to Reviewer wWdG**
>
> Many thanks for your detailed review. We address your comments below.
>
> 1) “Making inducing inputs shared parameters”:
>
> We do not argue that sharing inducing locations is necessary to avoid forgetting contributions from others, but rather if the inducing inputs are not shared parameters, then each client has its own set of say $M$ inducing points. In which case, when aggregating, the number of inducing points in the global sparse GP approximation will be $KM$. The computational complexity is then $O(K^3M^3)$ which, for a large number of clients $K$, quickly becomes computationally infeasible. We explain this in Section 5.
>
> 2) “Local neighbourhood of the training data”:
>
> We are unsure the exact point being made here, so apologies if this does not address your intended question. We could indeed combine local and global inducing point approximations in a manner similar to Snelson and Ghahramani (2007). However, this work is tangential to ours—if one were to achieve this in the federated learning setting, then our approach would be needed to aggregate client contributions to the local and global sparse GP approximations.
>
> 3) “Local clients should be allowed to customise their inducing locations”:
>
> As addressed in our response to 1), the principle issue with this is that it necessitates a cubic computational complexity in the number of clients when combining client contributions. An alternative approach, which we address in the related work section, is to share a “feature extractor” across client, but fit sparse GP approximations locally to each client (i.e. approximate the local posterior $p(f | D_k)$). If a global approximation is required, however, this approach is not suitable.
>
> 4) “Degenerated kernel”:
>
> We can indeed express the sparse GP as an equivalent exact probabilistic model with a degenerated kernel. However, the two interpretations do not result in equivalent optimisation approaches: if the interpretation of a degenerated kernel is taken, and ML is used to learn the parameters, then one recovers the expectation propagation (EP) approach to learning sparse GPs (see Bui et al. (2017) for a detailed discussion). When we introduce the HVM with the choice of auxiliary distribution given in Equation 6, it is interesting that we do indeed recover an equivalent to using a degenerate kernel with a prior over the inducing locations. We discuss this in Section 5.2.
>
> 5) “Decouple inputs mean and covariance parameters”:
>
> To clarify: our goal is not to decouple the inducing inputs from the mean and covariance parameters, but rather develop a method for performing federated learning of sparse GP approximations. The approach in Titsias and Lazaro-Gredilla is significantly different to the DPO parameterisation we develop in our paper, owing to different motivations: theirs is to enable the kernel hyper parameters to be integrated out, whereas ours is to reduce the stochasticity introduced into $q(u | Z)$ sampling the inducing inputs.
>
> 6) “Potential baseline”:
>
> In the above work, the $q(u)$ they use is the one that maximises the variational lower bound, whose parameters depend explicitly on the true inputs and observations and so is unsuitable for the FL setting. As we describe in Section 5.1, this is precisely why we introduce the alternative DPO parameterisation.
>
> 7) “Not very well motivated”:
>
> The motivations for our work are included in the introduction, namely that sparse GPs are effective in many domains, and there exist domains in which federated learning constraints are present, thus federated learning of sparse GPs is important. We have included additional citations in the revised edition.
>
> 8) “Will it perform on par with state-of-the-art, non-GP FL methods”:
>
> This depends on both the metric and task used to evaluate performance. In terms of accuracy, probably not; however, we expect that GP-based methods will outperform other approaches in terms of uncertainty metrics on low to medium dimension regression and classification tasks. Having said this, combinations of deep learning and GPs are known to be state-of-the-art on large-scale problems (see e.g. Liu et al. (2020, https://arxiv.org/abs/2006.10108)), and we anticipate our method being of use for some of these models. Nonetheless, we choose not to compare the performance of sparse GPs to other models in the tasks considered here as we consider this to be beyond the scope of the paper. Concretely, we have assumed a priori that a) sparse GPs are effective (see e.g. Bui et al. 2016, https://proceedings.mlr.press/v48/bui16.html for demonstration of the effectiveness of sparse GP-based models on small to medium sized regression tasks) and that b) there exist settings for which sparse GPs are desirable and the constraints of federated learning apply.

---

### Review · Reviewer_ANd7 · 2024-01-31

**Summary Of Contributions:**

The paper focuses on adapting Partitioned Variational Inference (PVI) for federated learning in the context of sparse Gaussian Processes. It addresses the challenge of learning variational parameters, particularly the inducing point locations, in a federated environment where data privacy is crucial. The authors propose a novel hierarchical variational model (HVM) framework to allow shared variational parameters across client-specific factors. This approach enhances communication efficiency and learning quality. The paper also introduces a new parameterization for variational distribution, improving the efficiency and quality of federated learning for sparse GPs. The contributions are validated through experiments on real-world regression tasks.

**Audience:**

Yes

**Claims And Evidence:**

No

**Requested Changes:**

Beside the weakness, many definitions of the symbols should be clarified.
E.g. N_k was not defined on Pg 3.
The reason why the variational parameters should be shared across factors should be clarified. I don't see the motivations here.
What does it mean by "free-form parameters"?
Eq. (3) is not equivalent toe Eq. (4).
what is tilda p in eq. 6?
What is the additional KL term mentioned in Eq (6) on Pg 4?
If alpha has to be 1 to ensure that PVI converges to VI, then why introducing it? Is that suggesting solution from VI is not better than PVI's?
What is V_k?
What is M_k?
What is the difference between X and y?

**Strengths And Weaknesses:**

Strengths:
- Innovative approach to extend PVI for sparse GPs with shared variational parameters.
- Development of a novel parameterization that may improve communication efficiency and learning quality.

Weakness:
- The paper may benefit from a deeper exploration of the insights of their approach. PVI was already proved to be efficient than FedAVG in Bayesian learning, so tailing FL with AVG and Sparse GP is a natural and special case of using the methodology of PVI. I don't see very clear contribution from this paper.
- Additional comparative analysis with other federated learning approaches could strengthen the findings.
- The paper is hard to follow. Paper writing could be significantly improved.

---

> ### Comment · Reviewer_ANd7 · 2024-03-13
>
> Since the authors didn't submit the revised version, which yet provided any useful feedback, I would suggest rejection.

---

> > ### Author Response · Authors · 2024-03-13
> > **Apologies. Rebuttal and revised paper to be posted today.**
> >
> > Please do accept our apologies for taking a while to post our responses to these comments, alongside a revised paper. We shall ensure that these are posted by the end of today.

---

> ### Author Response · Authors · 2024-03-13
> **Response to Reviewer ANd7**
>
> Many thanks for your review. We have addressed your concerns below.
>
> 1) “I don’t see any clear contribution from this paper”.
>
> We believe that there is some confusion here over what the contributions of our paper are. “tailing FL with AVG” is not our proposed approach—indeed, we highlight it as a naive approach one could take that we are explicitly seeking to outperform. Application of PVI is not natural, owing to the reasons discussed throughout the paper (see Section 1 paragraph 2, Section 4 paragraph 1, Section 5 paragraph 1). Namely, whilst PVI supports federated learning of model parameters that are shared across clients (in the SGP case, this is the function $f$), it does not support federated learning of variational parameters that are shared across clients (in the SGP case, this is the inducing locations).
>
> 2) “Additional comparative analysis with other federated learning approaches could strengthen the findings”.
>
> Other federated learning approaches are not suitable for learning sparse GPs, and we choose not to compare the performance of sparse GPs to other models in the tasks considered here as we consider this to be beyond the scope of the paper. Concretely, we have assumed a priori that a) sparse GPs are effective (see e.g. Bui et al. 2016, https://proceedings.mlr.press/v48/bui16.html for demonstration of the effectiveness of sparse GP-based models on small to medium sized regression tasks) and that b) there exist settings for which sparse GPs are desirable and the constraints of federated learning apply.
>
> 3) “The paper is hard to follow”.
>
> We apologise for this, and have updated / clarified more detailed parts of our paper in the revised edition. We suspect that this confusion arises because, although this paper falls within the scope of federated learning, its key contributions are more heavily within the field of variational inference (VI) and sparse GPs.
>
> 4) “Requested changes”.
>
> “N_k was not defined”: We have defined $N_k$ in the revised edition. It denotes the number of datapoints of client $k$.
>
> “Reason why variational parameters should be shared cross factors should be clarified”: The variational parameters that are shared across factors in sparse GPs are the inducing locations—each factor defines an unnormalised distribution over the function outputs at inputs $Z$ ($u = f(Z)$), thus they are dependent on the same variational parameter ($Z$).
>
> “I don’t see motivations here”: The motivations for our work are included in the introduction, namely that sparse GPs are effective in many domains, and there exist domains in which federated learning constraints are present, thus federated learning of sparse GPs is important. Further, there exist other models for which, when used in the federated learning setting, variational parameters are shared across clients. This includes variational autoencoders, for which the shared variational parameters are those of the encoder.
>
> “Free-form parameters”: We have rephrased the sentence that included the term “free-form parameters” (just below Equation 3) in the revised version. We hope that this provides additional clarity.
>
> “Equation 3 is not equivalent to Equation 4”: We are slightly confused as we make no claim that these equations are equivalent—the first is conditioned on $\phi$, and the latter is not.
>
> “What is tilda p”: We have provided additional clarity on this term in the revised version. Namely, this can be thought of as a “pseudo-prior” over the variational parameters, which we use to contruct a suitable PVI algorithm for HVMs.
>
> “Additional KL term”:  this KL term is a simplification of the KL term in Equation 5 using the choice for $r(\phi | \theta)$ defined just above Equation 6.
>
> “Why do we introduce alpha”: as mentioned in the paper, we find that $\alpha < 1$ often results in improved performance of the variational approximation, similar to its use in posterior tempering.
>
> $V_k$ are part of the `pseudo-observations’ for client $k$, as described beneath Equation 12.
>
> $M_k$ is the number of `pseudo-observations’ for client $k$.
>
> $X$ are inputs and $y$ are observations.

---

### Review · Reviewer_MLKV · 2024-02-11

**Summary Of Contributions:**

The paper develops an extension of partitioned variational inference (PVI) (Ashman, Bui et al; 2022) to address the problem of shared inducing locations across client-specific factors in a federated learning set up. The driving idea is to apply an hierarchical variational model (HVM) strategy, such that pseudo-observations are decoupled. The work initially introduces the problem of PVI in a general context, which is later explicitly framed for sparse Gaussian processes. The final advantage of the method is to use approximate likelihood terms in PVI, where each client k is parameterized by the set {$V_k, m_k, S_k$}. Experimental results show some advantages of using this sort of parameterization.

**Audience:**

Yes

**Claims And Evidence:**

Yes

**Requested Changes:**

If possible, I think addressing completely or partially some of the 4 issues raised above these lines in # weaknesses could help to improve my consideration of the manuscript.

**Strengths And Weaknesses:**

# Strenghts.
Good revision of PVI and good writing style. There are not technical typos or at least I have not detected any of them. Both formulation, modeling and methodology are well-developed and easy to follow, which is a strength. Experimental results are sufficient for the considered problems and for the length of the submission, with different goals and results to show. That's also positive. Both intro and discussion accurately describe what is actually done in the paper, which makes clarity higher.

# Weaknesses.
In general, I detect the following issues that I think should be considered for improvement.

**1) Novelty, federated learning, and PVI ideas:** The paper follows a similar structure as PVI, where the general partitioned variational inference framework is described first for later introducing sparse GPs (making a direct link between variables, for instance, $\theta$ with pseudo-observations $u$ and $\phi$ with inducing point locations $Z$). In that sense, this is very similar to the first version of PVI (not Ashman, Bui et al; 2022 in *arXiv:2202.12275* submitted in 2022, but *arXiv:1811.11206* from 2018). In that sense, some of the problems raised in the work about the lack of fitting of inducing point locations Z with the federated learning setting were somehow mentioned.

Even if this might not be entirely a critical problem, I find the DPO contribution somehow "small" in the paper, while PVI is re-introduced from the general VI view for later being re-adapted for GPs. This limits quite a bit the shine of the new ideas and contributions of the work. In my opinion, the analysis and ideas behind DPO are not explained in a careful way, apart from linking it to the HVM of Ranganath et al. ICML 2016. Additionally, as the authors correctly point out, the sort of parametrization used in Eq. 11 has been already considered in the recent GP literature, which also reduces the novelty of proposed ideas in the PVI context (in my opinion).

**2) Missing references in the context of sparse GPs:**

The first sentence in Section 5.2. is somehow an issue: "As far as we are aware, the only other work that utilizes probabilistic treatment of inducing locations Z is the Bayesian FITC method (Rossi et al., 2021)." One example in my mind is Jafrasteh et al. (ICML 2022) which explicitly introduces a distribution p(Z) in section 3 and a similar integral as the one considered in this submission can be found in Eq. (5). In general, I think that the probabilistic treatment of Z had been previously considered in the GP community, so to me that sentence is not 100% correct.

Even if the work develops an extension of PVI, I find other published works that have been omitted or not considered as key references, when they share key similarities or related problems. One example of this is Moreno-Munoz et al. (NeurIPS 2021), where some of the problems faced in the submission were already considered (i.e. having pseudo-observations per client, or non-global variables that are thus not affected by the federated learning framework). In the same way that here the set {$V_k, m_k, S_k$} is established per k-th client, a similar dictionary of variables is described in the paragraph before section 2.2. Also, the paper considers a similar problem using variational inference.

**3) Limited DPO analysis and explanation:** In my opinion, the section 5.1, which is the main technical contribution of the work is somehow small and not dense in useful details and justification of ideas, which makes the reader difficult to perceive why decisions are taken.

**4) Conflict with Federated Learning "traditional ideas":** Federated learning has became a rich area of research, with a particularly active sub-community around GPs. In FL, one of the primary goals, apart from the computational one is privacy preservation. I feel that the title of the paper, even being focused on PVI, does not discuss or models around the FL goals or ideas, which is somehow odd since FL is actually in the title of the submission.

# References:

Jafrasteh et al. (ICML 2022) > Input Dependent Sparse Gaussian Processes

Moreno-Munoz et al. (NeurIPS 2021) > Modular Gaussian Processes for Transfer Learning.

---

> ### Author Response · Authors · 2024-03-13
> **Response to Reviewer MLKV**
>
> Many thanks for your highly informative and useful feedback. We address each of your concerns below.
>
> 1) *Novelty, federated learning, and PVI ideas*:
>
> We would like to emphasise that our main contributions in this paper are twofold: 1) the introduction of ideas from HVMs to handle shared variational parameters across factors in PVI, and subsequent application to sparse GPs; and 2) the introduction of the DPO parameterisation to provide careful treatment of the additional stochasticity introduced by sampling the inducing point locations. The DPO parameterisation is not required for application of our method to sparse GPs, but it improves the performance of the variational approximation considerably.
>
> Equation 11 has indeed been considered in GP literature; however, this is not the DPO parameterisation. The DPO parameterisation is provided in Equation 12, which has not been considered in previous GP literature. We have provided some additional clarity on this in the revised paper.
>
> We are unsure as to what you mean by “some of the problems raised in the work about the lack of fitting of inducing point locations $Z$ with the federated learning setting were somehow mentioned.”
>
> 2) *Missing references in the context of sparse GPs*:
>
> Thank you for pointing out the use of stochastic inducing locations in Jafrasteh et al. (ICML 2022). We have included a discussion on this in the related work section in the revised edition. In their case, the distribution over inducing locations is defined implicitly by passing mini-batches of inputs through a neural network. It is worth highlighting that this differs substantially to our approach, in which we maintain an explicit distribution for which we perform inference.
>
> Many thanks also for pointing Moerno-Munoz et al. (NeurIPS 2021). We have included a reference to this paper in the related work. Indeed, this paper formed an initial motivation for exploring how PVI could be applied to sparse GP approximations, as the approach of “variationally merging” multiple sparse GP with different inducing locations into another sparse GP with separate inducing locations is not suitable for PVI as the cavity distribution cannot subsequently be computed.
>
> 3) *Limited DPO analysis and explanation*:
>
> We slightly expanded Section 5.1 in the revised version to provide some additional intuition into how the DPO parameterisation differs from similar existing parameterisations.
>
> 4) *Conflict with Federated Learning “traditional ideas*:
>
> We have included an additional paragraph in the related work section of the revised paper which discusses more general federated learning. We have also included a discussion at the end of Section 3 which highlights some of the important similarities and differences between PVI and FedAvg, including a comment that PVI can be made DP through DP-PVI (Heikkila et al. 2022).

---

### Author Response · Authors · 2024-03-13
**Summary and Revised Paper**

We thank all reviewers for the time and effort put into their responses, and have used their feedback to update our paper (see attached). We would like to highlight that, although this paper indeed falls within the scope of federated learning, the technical insights and novelties are largely VI and sparse GP based. We believe that this has contributed to a significant degree of confusion and, whilst we address each of the reviewers concerns below, we hope that this does not deter from the consideration of acceptance.

We have updated the paper with a revised version, with the main changes being: 1) additional related work sections on federated learning and probabilistic treatment of inducing locations in sparse GPs; 2) a more detailed discussion on the relationship between PVI and traditional federated learning algorithms at the end of section 3; 3) a refined discussion of the inadequacies of PVI for variational approximations in which some variational parameters are shared across clients just after equation 3; and 4) an emphasis on the difference between the DPO parameterisation we develop and other, similar parameterisations developed by others. Other changes are addressed in our responses to reviewers.

---

### Decision · Action_Editor_KWqy · 2024-04-09

**Recommendation:** Reject

**Comment:**

Dear authors,

All the reviewers raised concerns with the initial manuscript. Though it appears that some of these issues could have been resolved with a revision, we have yet to receive an updated version. The current PDF seems identical to the original submission. Given that we are well beyond the review schedule, I have decided to reject the manuscript in its current form. However, I would like to encourage the submission of a revision in the future.

Although the reviewers' comments could likely be addressed quickly, they suggested that further elaboration and insights could be beneficial in the sections that describe this work's novel contributions. I would encourage incorporating these recommendations.

**Audience:**

Yes. The submission presents a framework for performing partitioned variational inference with shared variational parameters. The reviewers commended the interesting approach and thus, the submission is relevant for parts of the TMLR audience.

**Claims And Evidence:**

This paper introduces a framework for performing parallel variational inference in a federated learning setting. Various design choices are elaborated upon, and the method is presented in detail. Numerical experiments on synthetic data showcase the working mechanisms of the method. In these aspects, the contribution meets the acceptance criteria.

However, the reviewers recommended discussing and contrasting the proposed method with additional literature and conducting a comparative analysis with other federated learning approaches. Including this analysis will provide more robust evidence better to judge the overall approach relative to other baselines. In this sense, the evidence presented in the paper is not yet entirely convincing and clear.

While this is a borderline decision, I believe the benefits of addressing the reviewers' concerns in a revision are significant enough to warrant one in this case.